# Modern Metaproteomics: A Unique Tool to Characterize the Active Microbiome in Health and Diseases, and Pave the Road towards New Biomarkers—Example of Crohn’s Disease and Ulcerative Colitis Flare-Ups

**DOI:** 10.3390/cells11081340

**Published:** 2022-04-14

**Authors:** Céline Henry, Ariane Bassignani, Magali Berland, Olivier Langella, Harry Sokol, Catherine Juste

**Affiliations:** 1PAPPSO, Micalis Institute, INRAE, AgroParisTech, Université Paris-Saclay, 78350 Jouy-en-Josas, France; celine.henry@inrae.fr (C.H.); ariane.bassignani@gmail.com (A.B.); 2Micalis Institute, INRAE, AgroParisTech, Université Paris-Saclay, 78350 Jouy-en-Josas, France; harry.sokol@aphp.fr; 3MGP, INRAE, Université Paris-Saclay, 78350 Jouy-en-Josas, France; magali.berland@inrae.fr; 4PAPPSO, GQE-Le Moulon, AgroParisTech, CNRS, INRAE, Université Paris-Saclay, 91190 Gif-sur-Yvette, France; olivier.langella@universite-paris-saclay.fr; 5Gastroenterology Department, INSERM, Centre de Recherche Saint-Antoine, CRSA, AP-HP, Saint Antoine Hospital, Sorbonne Université, 75012 Paris, France; 6Paris Centre for Microbiome Medicine (PaCeMM) FHU, Service de Gastro-Entérologie et Nutrition, Hôpital Saint-Antoine, Direction de la Recherche Clinique et de l’Innovation (DRCI) de l’AP-HP, CEDEX 12, 75571 Paris, France

**Keywords:** metaproteomics, mass spectrometry, biomarkers, Crohn’s disease, ulcerative colitis

## Abstract

Thanks to the latest developments in mass spectrometry, software and standards, metaproteomics is emerging as the vital complement of metagenomics, to make headway in understanding the actual functioning of living and active microbial communities. Modern metaproteomics offers new possibilities in the area of clinical diagnosis. This is illustrated here, for the still highly challenging diagnosis of intestinal bowel diseases (IBDs). Using bottom-up proteomics, we analyzed the gut metaproteomes of the same twenty faecal specimens processed either fresh or after a two-month freezing period. We focused on metaproteomes of microbial cell envelopes since it is an outstanding way of capturing host and host–microbe interaction signals. The protein profiles of pairs of fresh and frozen-thawed samples were closely related, making feasible deferred analysis in a distant diagnosis centre. The taxonomic and functional landscape of microbes in diverse IBD phenotypes—active ulcerative colitis, or active Crohn’s disease either with ileo-colonic or exclusive colonic localization—differed from each other and from the controls. Based on their specific peptides, we could identify proteins that were either strictly overrepresented or underrepresented in all samples of one clinical group compared to all samples of another group, paving the road for promising additional diagnostic tool for IBDs.

## 1. Introduction

Intestinal Bowel Diseases (IBDs) represent a set of chronic inflammatory disorders of the intestinal tract, affecting more than 2 million Europeans and 1.5 million North Americans, increasing in incidence worldwide, and being a serious issue of public health [1,2]. IBDs include Crohn’s disease (CD) and ulcerative colitis (UC), two chronic progressive inflammatory diseases that develop in relapses, and are regarded as an excessive immune response of genetically susceptible hosts to commensal microorganisms [3]. Both have overlapping symptoms, such as diarrhoea, significant weight loss, abdominal pain, and extra-intestinal manifestations [3], yet they affect the digestive tract in different ways. Ulcerative colitis is characterized by superficial and continuous ulcerations of the colonic mucosa. In the period of remission, the mucosa may look normal. In flare-ups, the clinical phenotype is heterogeneous, which makes this disease difficult to diagnose [4]. Crohn’s disease is also characterized by ulcers, but also intestinal fistulas. Unlike UC, the inflammation is discontinuous, and all parts of the intestine can be affected, although it is more common in the terminal ileum. Inflammation is transmural, which means that it affects different layers of the intestinal wall, while UC is limited to the mucosa [5]. Importantly, 70–75% of CD patients require bowel resection when symptoms become life-threatening (intestinal perforation, refractory bleeding) or refractory to medical treatment, when only 25–30% of UC patients undergo surgery [3,4].

Currently, faecal calprotectin, a cytosolic protein released by activated neutrophils in response to inflammation, can be used as a cost-effective, non-invasive test to identify an inflammatory bowel condition, although it is not a specific marker. However, the correct diagnosis of these diverse IBD phenotypes enables the clinician to adapt the pharmacological treatment specific to each condition and its severity [3], and the earlier the diagnosis is made, the better the chance of avoiding the development of irreversible injuries. Today, the diagnosis of IBD is based on the combination of clinical symptoms and objective findings from blood tests, endoscopic, histologic, and imaging procedures (computerized tomography scan, magnetic resonance imaging) [3]. Despite these efforts, some patients will have to wait months or even years before putting a name on their condition. This prompts us to explore new diagnosis tools, preferably non-invasive, to assist in the diagnosis of IBDs.

Although the causes of IBDs are poorly understood, connections with the gut microbiota is now clearly established. Indeed, microbiota is required for the development of inflammation in genetically predisposed colitis animal models [6], reinfusion of luminal contents after ileal resection rapidly produces recurrent disease in Crohn patients [7], antibiotics such as metronidazole have been shown to delay postoperative recurrence of CD [8], and the hitherto identified susceptibility polymorphisms contribute to, or have plausible functional connections with bacterial sensing through innate and adaptative immune pathways including autophagy [9,10,11]. Clearly, the host microbiome interaction could be a triggering factor for IBD [12]. So, in addition to standard treatments with immunosuppressants [13,14] that aim at reducing inflammation, it is only natural to search for signals of the disease within the gut microbial community as well seek to correct any imbalance by the regular ingestion of classical [15,16] or next-generation [17] probiotics, prebiotics [18], postbiotics [19], or even more drastic interventions, such as faecal transplantation [20]. A relatively huge number of studies (Appendix A) have focused on microbial populations, originally and still now by 16S rRNA gene sequencing, and more recently by shotgun metagenomics, providing images of who are there and what they can potentially do, provided they are still alive. Much less is known about the true activity of the diverse living microbial members in the disease compared to healthy condition. Metaproteomics is the method of choice to answer this question, to identify functionally active microbial members and their products that are specific to a clinical condition and could serve as surrogate markers of the disease. Such studies are still under-developed (Appendix A), but progress in combined liquid chromatography-mass spectrometry, availability of reference databases from metagenomic sequencing, bioinformatic tools and huge computing resources clearly boost this emerging field. Pioneering proof-of-concept studies [21] and a handful of more classical studies [22,23,24,25,26,27] have already highlighted IBD-specific proteins and pathways that were unsuspected based on DNA approaches. This leads us to believe that the gut metaproteome is really emerging as the necessary complement of the gut metagenome, to make headway in understanding and combating dysfunctions of the microbiome in IBDs and other pathological states. Several reviews strongly promote this idea [28,29,30], which is actively encouraged by the recently created Metaproteomics Initiative [31,32] and efforts to provide benchmarks for gut metaproteomics [33].

One of the main problems in IBD is to differentiate between the different IBD phenotypes (CDC for Colonic Crohn’s disease, CDIC for Ileo-Colonic Crohn’s disease and UC for Ulcerative Colitis). We wanted to probe the potential of modern gut metaproteomics for highlighting proteins whose abundance differed between these patients, as well as identify the organisms they came from and the functions they supported. Importantly, we did not filter out human proteins that were tightly attached to microbial surfaces, as they can be a valuable source of information on the host’s condition, and a unique opportunity to discover unsuspected biomarkers, as shown in the present paper. Our pilot study shows the strength of modern metaproteomics for identifying which members are really operating and what they do in an inflammatory episode of CD or UC patients, thus demonstrating the power of this approach for assisting in the diagnosis of CD or UC flare-ups at hospital admission with a simple stool collection.

## 2. Materials and Methods

### 2.1. Volunteers and Sample Collection

We conducted a cross-sectional study including twelve patients with active Intestinal Bowel Disease (IBD; ten women and two men, aged 24 through 51 years) and eight healthy controls (CTRL) matched for age, sex, and weight. Patients were followed and hospitalized in the Hepato-Gastro-Enterology Department of the Saint-Antoine hospital (Paris). We made a rigorous selection of different phenotypes for this pilot study: seven patients were diagnosed for an active ulcerative colitis (UC), and five patients for an active Crohn’s disease (CD), either with ileo-colonic (CDIC, *n* = 2) or exclusive colonic (CDC, *n* = 3) localization. Exclusion criterion was the use of antibiotics within the preceding 2 months, but all patients were treated with either salicylic derivatives, or immunosuppressants, or anti-TNF or monoclonal antibodies, or a combination of these therapies. The control group comprised healthy volunteers with neither symptoms nor a family history of gastrointestinal disease, and with no use of medication. All individuals belong to the Suivitheque study and were recruited in the Gastroenterology Department of the Saint Antoine Hospital (Paris, France) and provided informed consent. All samples were obtained between October 2017 and August 2018. Approval for human studies was obtained from the local ethics committee (Comité de Protection des Personnes Ile-de-France IV, IRB 00003835 Suivitheque study; registration number 2012/05NICB).

Every participant was asked to provide a single fresh stool sample collected in a Stomacher^®®^ 400 circulator standard bag (ref BA6141 from Seward Medical, West Sussex, BN14 8HQ, United Kingdom), which was left open in a one-litre hermetic plastic box containing a catalyst (Anaerocult^®®^ Ref 1.13829.0001 from Merck KGaA, 64271, Darmstadt, Germany) to generate anaerobic conditions. This faecal material was maintained in a cool box and transferred within 2 h into an anaerobic chamber (90% N_2_, 5% H_2_ and 5% CO_2_, Piercan Plastunion, 93 141 Bondy, France) for processing. The microbiota was extracted immediately from the fresh donations and the extraction was repeated from the same stool specimens that had been frozen for two months at −80 °C, in order to select markers that are valid in the case where the samples should be routed to a distant diagnosis centre.

### 2.2. Sample Preparation

Microbiota were separated from the faecal matrix by flotation in a preformed Iodixanol (OptiPrep™, Ref 1114542 from ProteoGenix, 67300 Schiltigheim, France) continuous gradient from about 1 g stool aliquots (either fresh or frozen-thawed), according to a variant of the method previously detailed [24]. Here, we just reduced the size of the gradients: 5 mL of a mixture of OptiPrep™/HEPES buffer, 15 mM in saline, pH 7.0 (Ref Sigma Aldrich PHG0001-100G, Merck), 1:2 (*v/v*), in Ultra-Clear™ tubes (Ref 344060 from Beckman Coulter France, 93420 Villepinte). Moreover, faecal dilutions (1 g of stools supplemented with 7.25 mL of faecal diluent composed of OptiPrep™/HEPES buffer, 40 mM in saline, pH 7.0, 3:1 (*v/v*)) were roughly filtered (Stomacher^®®^ 400 classic strainer bags, ref BA6041 from Seward Medical) before being loaded under the preformed gradients, to avoid any occlusion of the needle. During low-speed ultracentrifugation (Optima XPN-80 ultracentrifuge, Beckman Coulter France) in a swinging SW 40 Ti rotor (Beckman Coulter France, 14,567× *g*, 45 min, 4 °C), bacterial cells migrated up to their buoyant density (d 1.110–1.190) while the unwanted faecal matrix sedimented. After washing in cold HEPES (10 mM in saline, adjusted to pH 7.0), the extracted microbiota was frozen in liquid nitrogen then kept at −80 °C in 2 mL screw cap microtubes (Ref 72693005 from Sarstedt France, 70150 Marnay). For bacterial lysis, 1.5 mL of cold saline Tris-EDTA buffer (50 mM Tris-HCl, pH 7.8 containing 150 mM NaCl and 1 mM EDTA), extemporally supplemented with PMSF (Ref 93482-50ML-F from Sigma Aldrich; final concentration 2 mM) and protease inhibitor cocktail (cOmplete™, EDTA-free Protease Inhibitor Cocktail, Ref 04 693 132 001 from ROCHE, provided by Sigma Aldrich; final concentration of 1.3X), was directly added to each frozen bacterial pellet. The pellets were dispersed by vigorous vortexing and sonicated on ice using a VCX 500 ultrasonic processor equipped with a 3 mm diameter probe (Sonics Materials, distributed by Fischer Scientific, 67403 Illkirch, France) in short intervals of 10 sec ON/10 sec OFF, with 20% amplitude, and for two 5 min periods separated by a 15 min break on ice with periodic vigorous vortexing. Finally, the suspension was centrifuged at 500× *g* for 15 min at 4 °C to remove unbroken cells and large cellular debris. The supernatant was ultra-centrifuged in a swinging rotor (SW 55 Ti in an Optima XPN-80 ultracentrifuge, Beckman Coulter France) at 220,000× *g* for 30 min at 4 °C to separate cell envelopes (pellet) from the cytosolic fractions (supernatant). Only the envelope fractions were used in this study, as they stand as the first line of interaction with the host.

Purification and digestion of proteins were performed according to SOPs previously detailed [34] except that the trypsin enhancer surfactant ProteoaseMAX^TM^ was replaced by the non-ionic surfactant ALS-400 (Progenta™ Non-ionic Acid Labile Surfactant I, Ref ALS-400-5 from Protea Biosciences, Morgantown, WV 26505, USA). Forty microbiota LC-MS/MS analyses (twenty from freshly extracted microbiota and as many from postfreezing extractions) were carried out in a completely randomized design, with five additional well-distributed bulk samples and a blank between each injection.

### 2.3. LC-MS/MS Analyses

The analyses of peptides were obtained using an UltiMate^TM^ 3000 RSLCnano System (ThermoFisher Scientific, San Jose, CA, USA) coupled to an Orbitrap Fusion^TM^ Lumos^TM^ Tribrid^TM^ mass spectrometer (ThermoFischer Scientific, San Jose, CA, USA). Trypsic digestion products (5 µg) were loaded, concentrated, and desalted on a precolumn cartridge (stationary phase: C18 PepMap 100, 5 µm; column: 300 µm × 5 mm) and desalted with a loading buffer 2% ACN and 0.08% TFA. After 4 min, the precolumn cartridge was connected to the separating RSLC PepMap C18 column (stationary phase: RSLC PepMap 100, 3 µm; column: 75 µm × 500 mm). Elution buffers were A: 2% ACN in 0.1% formic acid (HCOOH) and B: 80% ACN in 0.1% HCOOH. The peptide separation was achieved with a gradient from 0% to 35% B for 160 min at 300 nL/min, then 50% B for 170 min at 300 nL/min. One run took 195 min, including the regeneration and the equilibration steps at 98% B. Peptide ions were analysed using Xcalibur 4.1.5 with the following data-dependent acquisition steps: (1) full MS scan (mass-to-charge ratio (*m*/*z*) 400 to 1600, resolution 120,000) and (2) MS/MS (HCDOT, collision energy = 30%, resolution 15 000). Step 2 was repeated in top speed mode with a cycle time equal to 3 s. Dynamic exclusion was set to 60 s. Mass data interpretation was carried out as detailed previously [34] following an iterative method. Three-step interrogation of the concatenated databases IGC 9.9, *Homo sapiens* Swiss-Prot-TrEMBL (release April 2018) and contaminant with an e-value threshold of 0.05 for peptides and proteins was achieved. A minimum of two distinct peptides identified across all samples in the dataset was set to validate a protein, in order to exclude proteins with weak proof of presence. The presence of a protein was attested if it contained at least one specific peptide, which is a peptide that is not seen in any other protein. The grouping of proteins into protein subgroups (also called ‘metaproteins’) was done as previously described using the grouping algorithm included in X!TandemPipeline [35,36] based on the principle of parsimony. Each protein subgroup contains all proteins which are identified by the same set of peptides, i.e., those proteins which are indistinguishable based on the observed peptides. For all identifications, four types of modifications were searched: carbamidomethylation of cysteines (fixed modification), oxidation of methionines, excision of the N-term methionine with or without acetylation, and cyclization of Nterm (potential modifications). The mass tolerance was set to 5 ppm for the parent peptide and 10 ppm for the fragments. One miscleavage was allowed.

### 2.4. Statistics

To compare protein number and abundance between clinical groups, we implemented Wilcoxon tests with Benjamini–Hochberg stepwise adjustment for multiple pairwise comparisons between clinical groups (adjusted *p*-value threshold = 0.05). Statistical computing and graphics were performed in the R environment.

### 2.5. Search for Contrasts

Abundance of proteins was approached by the sum of their specific spectral counts. Given the low number of individual faecal samples, we applied a highly stringent selection, only retaining those proteins that were either strictly overrepresented or underrepresented in all samples of one group compared to all samples of another group. An iterative strategy was applied, starting with the search for markers that distinguished between all IBD samples and all CTRL, then refining search for contrasts between the three IBD phenotypes.

### 2.6. Taxonomic and Functional Annotation

Annotations were as previously detailed [34]. Briefly, all proteins embedded within each microbial protein subgroup were taxonomically annotated with the sequence aligner DIAMOND (Double Index Alignment of Nextgeneration sequencing Data) against the nonredundant NCBI database, with an e-value threshold of 10^−4^. The complete taxonomic assignment (from superkingdoms to species) of the hit with the best bit-score was designated as the taxonomic assignment of the protein. Then only protein subgroups (about 90%) whose all component proteins shared the same taxonomic annotation at the species level were functionally annotated using the KEGG (Kyoto Encyclopedia of Genes and Genomes) resource (release 89.0), with an e-value threshold of 10^−5^, a bit-score threshold of 60, and using the sensitive mode of DIAMOND. The functional annotations with the better bitscores were assigned to the proteins. When multiple functions were assigned to the same protein, all of them were considered so that each protein subgroup was functionally annotated with all KEGG Orthology (KO) entries assigned to all its component proteins. The functional and taxonomical annotations of the IGC 9.9 database are available [37].

## 3. Results and Discussion

### 3.1. Metaproteomic Profiling of Stool Samples

Using the iterative interrogation of IGC 9.9 concatenated with the human proteome database, we identified a total of 231,500 peptides and 43,521 subgroups of proteins (or metaproteins) across all samples, with an a-posteriori peptide FDR returned by X!TandemPipeline of 0.30%, i.e., far below the 1% threshold commonly advocated.

We first compared the metaproteomic landscape of fresh and frozen samples, based on the abundances of all the 43,521 protein subgroups in the 40 samples. The correlation matrix in Figure 1a shows a strong correlation (r > 0.9) in all samples but two, between metaproteome profiles obtained from either fresh or frozen aliquots. The low correlation (r = 0.56) observed for sample S09 comes from contamination of the fresh bloody sample, but not the settled frozen one, by erythrocyte proteins that we could identify as highly abundant in the dataset of the fresh sample. Unsupervised clustering of samples confirmed that pairs of fresh and frozen samples were closely related (Figure 1b).

Figure 2 summarizes the diversity per sample within each group of subjects, revealing a loss of proteins, both overall and per taxonomic group (Figure 2a), and activities (ko number abbreviated as n.ko on Figure 2b), in the three patient groups, which, in addition, were much more heterogeneous than the controls (see Appendix A for the significance of all pairwise comparisons between clinical groups). This is in line with the well-known heterogeneity of IBDs. An exception concerned the functional diversity (n.ko on Figure 2b) of the CDIC samples which remained uniformly high and not significantly different from the controls due to a flurry of activity of *E. coli* (detailed in the following). Interestingly, we found that more than 90% of the bacterial protein subgroups could be robustly annotated at the species and functional level. Since we only accepted annotation of a subgroup when all its protein members shared the same annotation, this is a further illustration of the biological relevance of grouping algorithms such as X!TandemPipeline for clustering proteins based on peptide sharing rules [34,35,36,38]. As an additional point, proteins from human origin were more numerous in patients, notably in CDC and UC (Figure 2a with statistical report in Appendix A). They could not be discarded despite two successive washing steps of the microbial pellets, probably because of their high abundance and strong adherence to bacterial surfaces.

Figure 3 illustrates the diversity (number of circles) and abundance (size of circles) of proteins across the different taxonomic entities (species level in white, genus level in dark colour and phylum level in lighter colour), including proteins of human origin. Interactive, zoomable circlepackeR charts are available as Appendix A, and related statistics are in Appendix A. In the three IBD groups, we observed a decrease in the diversity and abundance of proteins from the phylum *Firmicutes*. Moreover, CDIC samples, by comparison with the other three groups, were characterized by an increase in proteins from *Proteobacteria*. Striking differences were also observed within phyla. For instance, proteins from the genus *Blautia* and then *Faecalibacterium* dominated the phylum *Firmicutes* in controls, CDC, and UC, but came only in fourth and twelfth position, respectively, in CDIC; proteins from the genus *Bacteroides* dominated the phylum *Bacteroidetes* in controls, CDC and CDIC, while proteins from the genus *Prevotella* were dominant in UC. A further example are proteins from the genus *Escherichia*, which dominated the phylum *Proteobacteria* in all IBD, but came only fifth in the controls. At last, proteins from *Akkermansia muciniphila* invaded the phylum *Verrucomicrobia* of CDIC samples (see interactive Appendix A for any detailed inspection). These figures, here first reported from a true functional perspective, are well in line with the few DNA-based studies which distinguished CDIC, CDC, and UC, and identified disease phenotypes and localization as a major determinant of the gut microbial composition in IBD [39,40], while others did not [41], perhaps as a result of the disease activity at the time of sampling. The diversity and abundance of human proteins were increased in the three IBD groups, and this was the most striking in the UC samples. The increase in the total abundance of human proteins did not reach significance for CDIC samples only (Appendix A). Interestingly, while human protein species were in a minority compared to bacterial proteins (Figure 2a), their contribution in terms of abundance was much higher (Figure 3). Chymotrypsin-C, IgGFc-binding protein, and Polymeric immunoglobulin receptor (bold characters in Appendix A) were in the top list of the most abundant human proteins in all groups. A subset of ten human proteins (red characters in Appendix A), including calprotectins and other serine proteases were in the top list of patients only. All of them are linked with inflammation, regulation of autophagy and innate immunity, or have an antimicrobial activity, and some have been proposed as potentially useful biomarkers of IBD activity [42,43]. Interestingly, we identified a group of 26 human proteins (purple characters in Appendix A) which were in very low abundance in all CTRL and CDIC samples, in high abundance in UC samples except patient S10 (already identified as a low-inflammation grade outlier on Figure 1b), and in variable abundance in CDC samples. A number of them are related to bactericidal defence of the host, which has, a priori, no reason to be oversecreted in healthy conditions, and would be deficient in CDIC patients as against in UC or even CDC patients. This suggests another way of looking at results, avoiding to neglect a number of similarities with controls, which could in fact be signatures of insufficient defense. In addition, these results well illustrate the value of interrogating both metagenomic and human data bases to interpret mass spectra in gut metaproteomics as an outstanding way of capturing host and host–microbe interaction signals that DNA-based studies cannot provide.

The immense interest of gut metaproteomics is to decipher activities of living microbes at a given point in time, here in different inflammatory bowel flare-ups. From the protein list annotated for taxonomy and functions, and the spectral count table normalized for the number of samples per clinical group, we can address any question relative to the true metabolic activities of diverse microbial communities or subcommunities (Appendix A). For instance, pathways of galactose metabolism (map00052), starch and sucrose metabolism (map00500), fructose and mannose metabolism (map00051), tyrosine metabolism (map00350), sulfur metabolism (mp00920), valine, leucine, and isoleucine biosynthesis (map00290), and glycolysis/gluconeogenesis (map00010) were all depleted (Appendix A), while pathways of oxidative phosphorylation (map00190), glyoxylate and dicarboxylate metabolism (map00630), glycine, serine, and threonine metabolism (map00260), valine, leucine, and isoleucine metabolism (map00280 and map00290), or some reactions of the methane metabolism pathway (map00680) were all exacerbated in *Proteobacteria* of all three clinical groups compared to controls (Appendix A). Looking at lower taxonomic levels leads to the recognition of more focussed outcomes. For instance, only microbiomes of CDIC samples were characterized by a flurry of activity of *E. coli*, notably fatty acid biosynthesis (map00061), lipoic acid metabolism (map00785), oxidative phosphorylation (map00190), purine (map00230) and pyrimidine (map00240) metabolism, and lipopolysaccharide biosynthesis (map00540), which are all crucial for survival and virulence of gram-negative bacteria (Appendix A). At the same time, loads of activities were collapsed in the three other main phyla, *Firmicutes* (Appendix A), *Bacteroidetes* (Appendix A), and *Actinobacteria* (Appendix A), in the three patient groups while fewer activities were increased (Appendix A).

### 3.2. Search for Metaproteomic Signatures Specific of IBD Phenotypes

As the protein profiles of faecal microbiota prepared from the same sample either fresh or frozen were closely related (Figure 1), we considered that the same candidate biomarkers could apply to fresh as well as frozen samples. Therefore, all searches for specific contrasts were performed on the pooled freshly and post-freezing prepared microbiota, giving unique lists of markers that can be useful for further developments of routine clinical tests for all types of samples. Importantly, given the low number of samples, we applied a highly stringent selection of candidate biomarkers and retained only protein subgroups that were strictly over- or underrepresented in all samples (either fresh or frozen) from a group of subjects compared to all samples from another group.

We first searched subgroups that may segregate controls and all IBD patients. We identified seven proteins from human origin which were systematically in a greater abundance in patients than in controls (Figure 4a). One of them (subgroup b43.a1) is the well-known neutrophil-derived protein S100-A9 predominantly found as the S100A8/S100A9 complex, also known as calprotectin [44,45]. Calprotectin is activated when inflammation occurs for many different reasons. Thus, when abdominal symptoms exist, the dosage of faecal calprotectin can be used to identify an inflammatory bowel condition and determine the next course of action in diagnosis and treatment. Importantly, faecal calprotectin is a useful and cost-effective marker to help differentiate between IBD and IBS (Irritable Bowel Syndrome), but does not differentiate between different IBD phenotypes, as also proved for the first time by our envelope-targeted metaproteomic approach. This result, retrieving a well-known and already widely used biomarker, validates our approach. Interestingly, we found another member of the S100 protein family, S100-A12, to be increased in all IBD patients compared to all controls. S100-A12 is known to be elevated in a range of inflammatory conditions, including IBD [46]. We also highlighted five additional neutrophil-derived proteins that were even more increased than calprotectin and S100-A12 in all patients compared to all controls, and whose robust identification, based on a large number of peptides, is disclosed in Appendix A (sheet 1). These are the proteins embedded in the azurophilic granules of neutrophils (myeloperoxidase, neutrophil elastase, cathepsin G, myeloblastin and azurocidin), which are all related to host defence against bacterial infection [47], some of them having been reported to support the differentiation of chronic IBD from IBS and correlate with the severity of IBD inflammation [48]. Therefore, our findings well reflect the release of a number of proteins by activated and degranulated neutrophils, which is consistent with the exacerbation of immune response in IBD. This clearly demonstrates that gut metaproteomics is a powerful tool for relevant marker confirmation or discovery, not only from bacterial, but also from human origin. Looking for proteins less abundant in patient samples revealed one protein from bacterial origin, identifed as the ATPF1B (ATP synthase subunit beta, K02112) of *Coprococcus comes* (family *Lachnospiraceae*), which could not be detected in any patient but was present at low levels in all controls (Figure 4a and sheet 1 of Appendix A). *C. comes* accounts for part of those commensals which are regularly reported as being less abundant in both UC and CD microbiomes, based on whole-genome shotgun sequencing of faecal DNA [49,50]. Moreover, increased levels of antibodies to *C. comes* have been reported to be more frequently found in sera of CD patients, leading to propose this serological test as an adjunct in the diagnosis of IBDs in the 1980s [51], which, however, never translated into useful clinical tools.

Interestingly, many additional differences were observed when we compared each clinical group to controls (see sheets 2–4 of Appendix A), offering a wealth of potential biomarkers to differentiate the diverse IBD phenotypes from a healthy condition. Clearly, CDIC microbiomes were characterized by a flurry of activity of *Escherichia* and other members of *Enterobacteriaceae*, together with the extinction of activity of *Faecalibacterium prausnitzii* and to a lesser degree of *Anaerostipes hadrus* and *Fusicatenibacter saccharivorans*. CDC microbiomes were characterized by the increased activity of diverse *Bifidobacterium bifidum* organisms, while UC microbiomes were characterized by an impressive number and quantity of human proteins coating the gut bacteria, including, in addition to the above named neutrophil-derived proteins, innate immunity proteins such as peptidoglycan recognition protein 1 and integrins, or antimicrobial peptides such as cathelicidin antimicrobial peptide. We also found evidence of decreased activity of *Anaerostipes hadrus* in CDC samples, and of *Fusicatenibacter saccharivorans* in UC samples, but no evidence of decreased activity of *Faecalibacterium prausnitzii* in CDC or UC samples.

A main purpose of the present research was to find proteins that could differentiate between flare-ups of different IBD phenotypes, i.e., between CD and UC, and ideally between CDIC, CDC and UC. As CDIC is the most serious IBD phenotype, we started with the selection of the proteins whose abundance was systematically increased or decreased in all CDIC samples compared to all CDC or UC samples. We could identify 101 protein subgroups that were specifically overabundant in CDIC samples (Figure 4b, detailed in Appendix A). Among these, 97 were from bacterial origin, essentially from the family *Enterobacteriaceae* (*n* = 64), and most specifically from *E. coli* organisms in the vast majority, while *E. fergusonii* was also well represented. The remaining thirty-three bacterial protein subgroups belonged to one of the eight families, *Lachnospiraceae* (*n* = 11, most from *Clostridium clostridioforme*), *Bacteroidaceae* (*n* = 9), *Clostridiaceae* (*n* = 3), unclassified *Clostridiales* (*n* = 3), *Ruminococcaceae* (*n* = 2), *Acetobacteraceae* (*n* = 1), and *Morganellaceae* (*n* = 1), or could not be reliably assigned since their protein members were annotated to organisms from different families (*n* = 3). The four human proteins that we found to be overabundant in all CDIC samples compared to other IBD samples were enzymes involved in the host lipid metabolic process (Appendix A). Lastly, 87 of these 101 abovementioned proteins also emerged when CDIC samples were compared to all other samples, including the controls. They are delineated above the horizontal line on Figure 4b and indicated in the last column of Appendix A. Interestingly, we found only one host protein, ribonuclease A F1 (RAF1), which was specifically less abundant in CDIC samples compared to CDC and UC samples. RAF1 is known to regulate intestinal epithelial cell survival in response to pro-inflammatory stimuli [52]. The detailed identification of these candidate signatures of CDIC is disclosed in Appendix A.

CDIC samples therefore differ from CDC and UC samples, or even from controls, by an invasion of bacterial proteins mainly from *E. coli*, and to a lesser extent from and *E. fergusonii* and *Clostridium clostridioforme*. The invasion of a number of opportunistic pathogens, such as *E. coli* and *C. clostridioforme* was already reported in CD patients, but it was based on metagenomic shotgun sequencing and the comparison did not distinguish between ileo-colonic (CDIC) and exclusive colonic (CDC) localization of Crohn’s disease [41,53].

We are now at the point where we can suspect an inflammatory bowel condition based on a group of seven abundant immune cell-derived proteins and where we can reasonably suspect a Crohn’s disease with ileo-colonic localization based on an invasion of protein entities from *E. coli* and *C. clostridioforme*. We still have to distinguish between CDC and UC. We identified six protein subgroups which were more abundant in all CDC samples compared to all UC samples (Figure 4c, detailed in Appendix A). Four of them were from *Faecalibacterium* species (referred to as a1.c148, b78.a1, c278.b28, and d3555.a1 on Figure 4c), and the two others were pancreatic chymotrypsinogens (referred as to c113.a1 and c113.a2 on the same figure). We found no protein specifically less abundant in all CDC samples compared to all UC samples. Therefore, UC and CDC phenotypes, although offering much fewer candidate biomarkers than CDIC, could still be distinguished based on *Faecalibacterium* proteins and their 23 unique specific peptides (Appendix A).

## 4. Conclusions

Based on a refined metaproteomic analysis, it is possible to provide a detailed taxonomic and functional landscape of all living and active members of the gut microbiome at a given point in time, here in different inflammatory bowel flare-ups, together with their coating human proteins if desired. Moreover, with a highly stringent selection of protein subgroups that were either strictly over- or underrepresented in all samples of one clinical group compared to all samples of another group, we can propose a theoretical algorithm based on modern metaproteomics, which could be explored further on large cohorts (Figure 5). For each selected candidate protein, the best representative (proteotypic) peptides can be extracted from the mass spectrometry data deposited to the MassIVE repository [54] with the data set identifier MSV000089099 and the ProteomeXchange identifier PXD032706, and used to develop multiplexed selected reaction monitoring assays allowing to precisely quantify and confirm these candidate biomarkers. If confirmed in large trials, they could represent useful adjuncts to clinical grounds and objective findings of radiological, endoscopic, and histological examination, for the early diagnosis of CDIC, CDC, and UC. The bottleneck for the study of large cohorts is undoubtedly the sample preparation, which here included an extraction of the microbiomes from the faecal matrix. However, rapid progress in combined liquid chromatography-fast scanning high resolution mass spectrometry, together with the implementation of data-independent acquisition (DIA) in shotgun metaproteomics, which allows to fragment all peptides in a sample, should in the near future enable an unprecedent and rapid access to the whole complexity of non-prepurified microbiomes.

## Figures and Tables

**Figure 1 cells-11-01340-f001:**
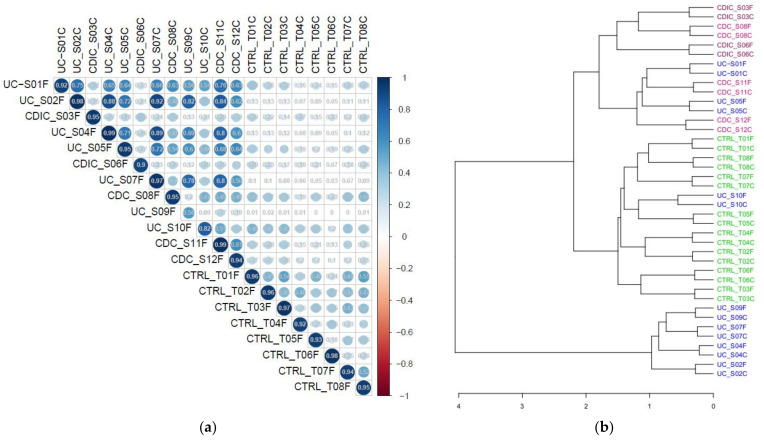
(**a**) Pearson’s correlation between metaproteomic profiles (abundance of each of the 43 521 subgroups) of fresh (suffixed with F) and frozen (suffixed with C) samples. (**b**) Clustering of samples based on their subgroup abundances. CTRL: controls (8 pairs of fresh/frozen samples); CDC: Colonic Crohn’s disease (3 pairs of fresh/frozen samples); CDIC: Ileo-Colonic Crohn’s disease (2 pairs of fresh/frozen samples); UC: Ulcerative Colitis (7 pairs of fresh/frozen samples). Samples UC_S10, which clustered with controls, corresponded with low inflammation.

**Figure 2 cells-11-01340-f002:**
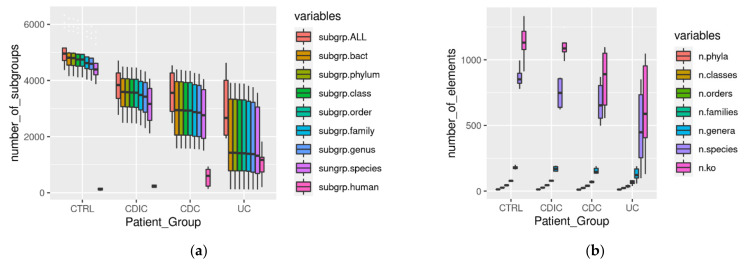
(**a**) Protein subgroups (abbreviated ‘subgrp’) per sample in each clinical group: subgr.ALL: total number of protein subgroups; subgrp.bact: total number of bacterial subgroups; subgrp.phylum to subgrp.species: total number of bacterial subgroups with consensual taxonomic annotation of all their protein elements from the phylum-to-species level; subgrp.ko: number of bacterial subgroups with consensual functional annotation of all their protein elements; subgrp.human: total number of human subgroups. (**b**) n.phyla to n.species: number of phyla, classes,… species per sample (only taxonomic annotations concordant within a same protein subgroup were counted); n.ko: number of KO entries per ample (only functional annotations concordant within a same protein subgroup were counted). Abbreviations and headcounts of patient groups are those detailed in legend of Figure 1.

**Figure 3 cells-11-01340-f003:**
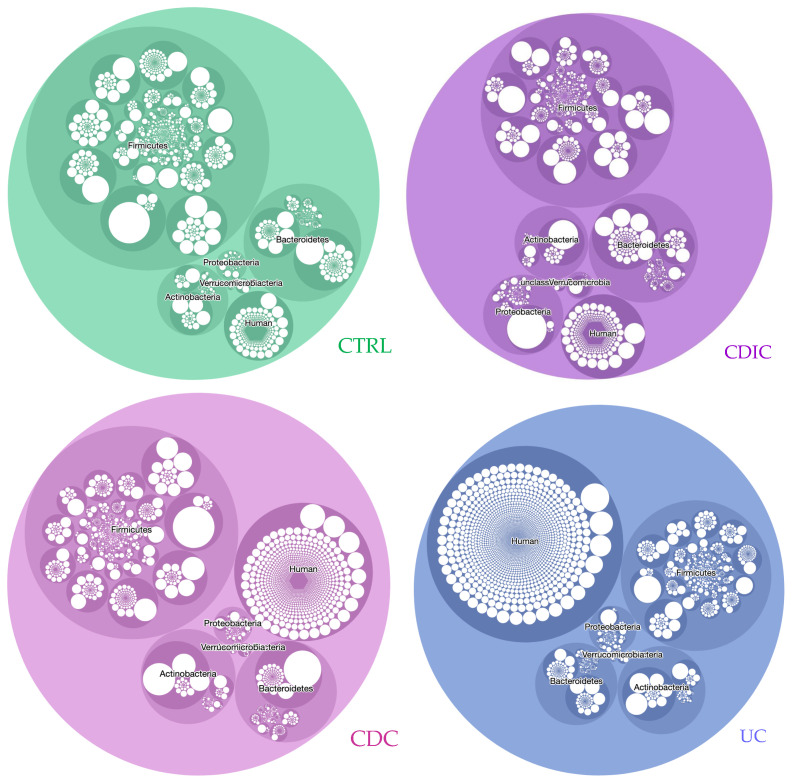
CirclepackeR graphs illustrating the intensity of true activities in the envelope-enriched fractions of the different taxonomic groups in CTRL and IBD microbiomes. Abundances of protein subgroups (computed as the sum of the spectral counts of their specific peptides) were summed within each taxonomic species, which were packed into genera and phyla. For readability, only the five main phyla, *Actinobacteria*, *Bacteroidetes*, *Firmicutes*, *Proteobacteria*, and *Verrucomicrobia*, plus unclassified bacteria and human proteins, are represented. See Appendix A for interactive detailed inspection. Abbreviations and headcounts of patient groups are those detailed in legend of Figure 1.

**Figure 4 cells-11-01340-f004:**
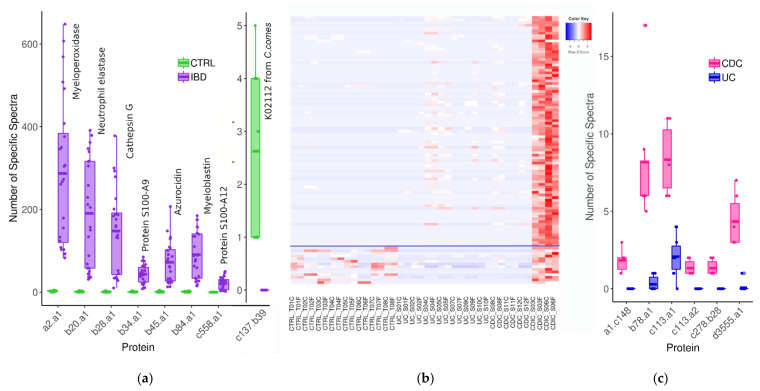
(**a**) Protein subgroups over- and underrepresented in all IBD samples compared to all controls. (**b**) Heatmap of 101 subgroups overabundant in CDIC samples compared to all other IBD samples (see Appendix A for their complete list); 87 of them (over the blue line) were also overabundant compared to controls (data are log-transformed and zero-centred). (**c**) Protein subgroups overabundant in all CDC compared to all UC samples. Abbreviations and headcounts of patient groups are those detailed in legend of Figure 1.

**Figure 5 cells-11-01340-f005:**
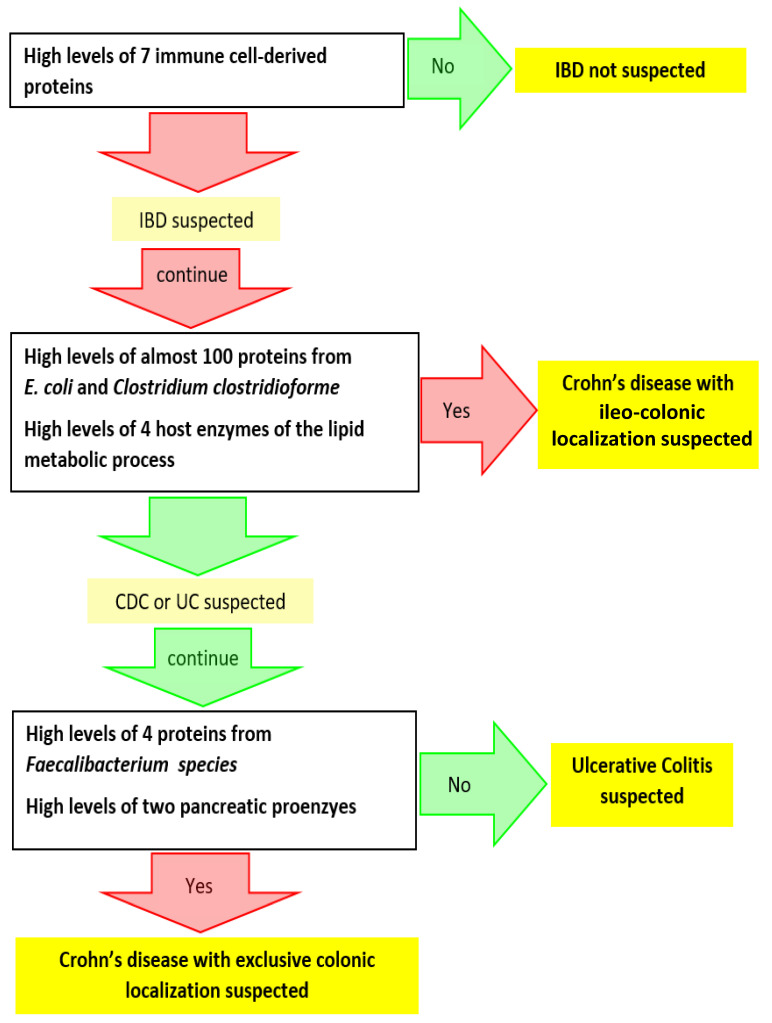
Leads for further trials aimed at consolidating a decision tree based on modern gut metaproteomics: a potential additional tool to supplement clinical grounds, medical imagery, and histology in the early diagnosis of IBDs.

## Data Availability

Raw data and results are deposited to the MassIVE repository [54] with the data set identifier MSV000089099, and the mass spectrometry proteomics data have been deposited to the ProteomeXchange Consortium [59] via the PRIDE partner repository [60] with the dataset identifier PXD032706.

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
