# Peer review of "Modern Metaproteomics: A Unique Tool to Characterize the Active Microbiome in Health and Diseases, and Pave the Road towards New Biomarkers—Example of Crohn’s Disease and Ulcerative Colitis Flare-Ups"

_cells, 2022, doi:10.3390/cells11081340_

Round 1

Reviewer 1 Report

It's good paper  I have no comments or suggestions

Author Response

We are very grateful to the Reviewer for his positive report.

Reviewer 2 Report

Manuscript ID: cells-1655395

Title: Modern Metaproteomics: A Unique Tool to Characterize the Active Microbiome in Health and Diseases, and Pave the Road 3 towards New Biomarkers. Example of Crohn’s Disease and Ulcerative Colitis Flare-ups

Authors: Céline Henry et al.

The above manuscript is interesting. The main achievement of the authors is the discovery that the results of the studies of microbial cell envelope metaproteomes from fresh stool samples and those frozen for a two month are similar. In addition, the authors found that the taxonomic and functional landscape of microbes in diverse IBD phenotypes, active ulcerative colitis, or active Crohn’s disease with ileo-colonic or exclusive colonic localization, differed from one another and from the controls. Moreover, the authors identified proteins that were either strictly over-represented or under-represented in particular forms of IBD. However, the number of observations is strongly limited and no data showing that the declared results are statistically significant.

In addition, it should be noted that the manuscript contains some shortcomings and some overinterpretations. For this reason, a major revision of the manuscript is necessary before accepting the manuscript for publishing.

Lis of errors:

  1. Introduction, page 1, line 39. The authors should write that IBD is a chronic and progressive inflammatory disease resulting from an inappropriate immune response, in genetically susceptible individuals, to microbial antigens of commensal microorganisms (PMID: 19568370). In addition, the authors should write more details on clinical features of CD and UC. Patients with Crohn’s disease often do not have bloody diarrhea (PMID: 19568370).
  2. Introduction, the second paragraph, line 52. The authors present methods used in diagnosis of IBD in the order of their current clinical utility (PMID: 19568370; PMID: 33885977). The diagnoses of Crohn’s disease and ulcerative colitis are made on clinical grounds supplemented with objective findings of radiological, endoscopic, and histological examination (PMID: 19568370).
  3. Introduction, the third paragraph. The authors should write some words on the role of probiotics, especially Escherichia coli strain Nissle 1917, in experimental models of IBD (PMID: 27433160), as well as clinical settings (PMID: 33361702).
  4. Introduction, line 94 and 113 and next lines. The authors should replace “colic” with “colonic”.
  5. Material and Methods, line 125 and next lines. In the case of each chemical, assay, kit, test or equipment, the authors should provide its generic name, name given by manufacturer, name of manufacturer or distributor, city of their headquarter, and name of the country.
  6. Material and Methods. The number of patients and healthy volunteers, as well as the number of observations was strongly limited. Did the authors perform any statistical analysis? It should be clearly described in the manuscript. It would be advisable to increase the number of observations so that conclusions are not based on speculation but on statistical analysis.
  7. Did the authors reach any statistically significant effect? This information should be shown in the manuscript.
  8. Tables and figures and their legends must be understandable without references to the content of the manuscript. All abbreviations in each figure should be presented in their full name in figure legends. The figure legend should clearly state what is shown on the Figure. All markers must be described in figure legend. Moreover, the exact number of observations in each experimental group should be shown in figure legends or on figures.
  9. Conclusion and Figure 5 are not supported by observations. It is clear speculation. No statistically significant results. No clear data showing that metaproteomic analysis of microbiota is useful in the diagnosis of IBD. Currently, metaproteomic analysis of microbiota may have some scientific value, but diagnosis of IBD must be made on clinical grounds supplemented with objective findings of radiological, endoscopic, and histological examination. This statement should be presented in the conclusion in the body of the manuscript and the abstract.

Author Response

We are grateful to the Reviewer for his constructive criticism that we have tried to make the best use to improve the manuscript. Our point-by point answers are joined below.

Reviewer 3 Report

The paper entitled “Modern Metaproteomics: A Unique Tool to Characterize the Active Microbiome in Health and Diseases, and Pave the Road towards New Biomarkers. Example of Crohn’s Disease and Ulcerative Colitis Flare-ups.” is a well written study reporting the potentialities of modern metaproteomics to improve the diagnosis of CD or UC with stool collection. The cross-sectional study included twelve IBD patients and eight healthy donors matched for age, sex and weight. Inclusion and exclusion criteria are reported as well as the number/ID of ethics approval. The work is well written and structured. The methods are clearly described and the results are of clear importance in the field. 

Author Response

(The authors gave the same response as above.)

Round 2

Reviewer 2 Report

The new, revised version of the manuscript is almost ready for publication. The occurring abnormalities are:

    Double use of the same article as reference 3 and 6.
    One type of English spelling should be used throughout the manuscript. The authors have changed “diarrhea” to “diarrhoea” in the present version manuscript. There are also other British spelling words such as centre, faecal, litre, which would indicate this type of spelling was chosen by the authors. However, it would be advisable to spell-check the final version of the manuscript.
    There should be no period at the end of the title.